# A Free Amino Acid Diet Alleviates Colorectal Tumorigenesis through Modulating Gut Microbiota and Metabolites

**DOI:** 10.3390/nu16071040

**Published:** 2024-04-03

**Authors:** Yang-Meng Yu, Gui-Fang Li, Yi-Lin Ren, Xin-Yi Xu, Zheng-Hong Xu, Yan Geng, Yong Mao

**Affiliations:** 1Department of Oncology, Affiliated Hospital of Jiangnan University, Wuxi 214122, China; 6212809077@stu.jiangnan.edu.cn (Y.-M.Y.); 6212809017@stu.jiangnan.edu.cn (G.-F.L.); 9862020004@jiangnan.edu.cn (X.-Y.X.); 2Department of Gastroenterology, Affiliated Hospital of Jiangnan University, Wuxi 214122, China; renyilin@jiangnan.edu.cn; 3School of Life Sciences and Health Engineering, Jiangnan University, Wuxi 214122, China; 4College of Biomass Science and Engineering, Sichuan University, Chengdu 610065, China; zhenghxu@scu.edu.cn

**Keywords:** free amino acid-based diet (FAA), colorectal cancer, intestinal flora

## Abstract

Colorectal cancer (CRC), a major global health concern, may be influenced by dietary protein digestibility impacting gut microbiota and metabolites, which is crucial for cancer therapy effectiveness. This study explored the effects of a casein protein diet (CTL) versus a free amino acid (FAA)-based diet on CRC progression, gut microbiota, and metabolites using carcinogen-induced (AOM/DSS) and spontaneous genetically induced (*Apc^Min/+^* mice) CRC mouse models. Comprehensive approaches including 16s rRNA gene sequencing, transcriptomics, metabolomics, and immunohistochemistry were utilized. We found that the FAA significantly attenuated CRC progression, evidenced by reduced colonic shortening and histopathological alterations compared to the CTL diet. Notably, the FAA enriched beneficial gut bacteria like *Akkermansia* and *Bacteroides* and reversed CRC-associated dysbiosis. Metabolomic analysis highlighted an increase in ornithine cycle metabolites and specific fatty acids, such as Docosapentaenoic acid (DPA), in FAA-fed mice. Transcriptomic analysis revealed that FAA up-regulated Egl-9 family hypoxia inducible factor 3 (Egln 3) and downregulated several cancer-associated pathways including Hippo, mTOR, and Wnt signaling. Additionally, DPA was found to significantly induce EGLN 3 expression in CRC cell lines. These results suggest that FAA modulate gut microbial composition, enhance protective metabolites, improve gut barrier functions, and inhibit carcinogenic pathways.

## 1. Introduction

Colorectal cancer (CRC) has emerged as a predominant global health concern, with both incidence and mortality rates escalating worldwide [1,2]. Particularly in developed regions such as Europe and North America, the prevalence of CRC is pronounced. However, this upward trend is equally alarming in many developing nations [3]. Notably, while CRC has traditionally been categorized as a disease of the middle-aged and elderly, there has been a discernible increase in its incidence among younger demographics in recent years [4,5]. Early prevention has thus become paramount. The progression of CRC often encompasses a sequential transition from normal mucosa to adenoma, culminating in malignant transformation [6,7]. This involves aberrant cell cycle regulation and the frequent activation of specific signaling pathways, such as the Wnt/β-catenin and TGF-β pathways, which facilitate tumor formation and advancement [8].

The integrity of the intestinal barrier plays a pivotal role in CRC development [9]. Its compromise allows for the easier ingress of external pathogens into the intestine, thereby accelerating CRC progression. In this context, the over-proliferation of specific pathogens like *Fusobacterium nucleatum* in CRC patients correlates with a decline in intestinal barrier function [10]. More crucially, the augmentation in the abundance of beneficial bacteria might be associated with their proteolytic activity on mucosal glycoproteins, thereby preserving the stability of the intestinal barrier, as seen with *Akkermansia muciniphila* [11,12]. Metabolically, long-chain polyunsaturated fatty acids (LC-PUFAs) are instrumental in cell membrane structure, signal transduction, inflammatory responses, and immune modulation. Especially, ω-3 fatty acids, such as Eicosapentaenoic Acid (EPA) and Docosapentaenoic acid (DPA), have been demonstrated to modulate gut microbiota equilibrium, enhancing the abundance of probiotics like *Lactobacillus* and *Bifidobacterium* [13].

Currently, while various therapeutic strategies for CRC exist, prolonged use may be accompanied by side effects. In contrast, dietary intervention approaches are more benign, with fewer adverse outcomes. In recent years, amino acid-rich diets have garnered extensive attention in the prevention and treatment of CRC [14]. Specific amino acids, such as arginine and ornithine, can bolster the proliferation and differentiation of intestinal epithelial cells, thereby maintaining the integrity of the intestinal barrier. They also influence the composition of the gut microbiota, with some studies indicating that arginine can promote the growth of emerging probiotics like *Akkermansia* and *Bacteroides* [15,16]. Additionally, these amino acids can modulate cancer-associated signaling pathways, such as mTOR, and regulate immune responses, particularly the function of T cells [17,18].

A free amino acid (FAA) diet consists of pre-digested proteins forming readily absorbable individual amino acids. This diet design aims to directly supply the nutrients required by the intestine, optimizing nutrient absorption and minimizing the digestive system’s load. Regarding the absorption ratio of free amino acids in the small intestine, studies indicate that the absorption rate of free amino acids is higher than that of intact proteins or protein hydrolysates, as they can be directly absorbed by the intestinal cells without the need for further digestion [19].

Although free amino acids offer these advantages, it is necessary to consider the potential adverse effects of aromatic amino acids (such as phenylalanine, tyrosine, and tryptophan). Long-term excessive intake of these amino acids may affect the balance and metabolism of neurotransmitters, sometimes leading to side effects such as mood swings, fatigue, or digestive disturbances. Therefore, when designing an FAA diet, these potential adverse reactions should be taken into account and adjustments made as necessary.

Recent studies found that microbiome-derived metabolites are derived from unabsorbed dietary protein that reaches the colon [20,21,22]. However, the intricate relationship between intact proteins or FAA and microbiome dysbiosis and its metabolite changes in CRC development remains largely unknown. Here, we aim to elucidate how FAA-based diets influence the gut microbiota and their impacts on the genetic and metabolic levels. Both chemically induced and spontaneous CRC mouse models were employed for dual validation. We discovered the potential interactions between gut microbiota and metabolic products, leading to the inhibition of specific pathways, thereby decelerating CRC progression.

## 2. Materials and Methods

### 2.1. Colorectal Cancer Mouse Models and Treatments

BALB/c mice (male, 6 weeks old) and *Apc^Min/+^* C57BL/6 mice (male, 6 weeks old) were purchased from GemPharmatech Co., Ltd. (Nanjing, China). The mice were housed in a specific pathogen-free (SPF) condition, with ad libitum access to food and water. All animal procedures were approved by the Institutional Animal Care and Use Committee of Jiangnan University (Approval number No: JN.No20220930m1200424[373]). For the carcinogen-induced CRC model (AOM/DSS), after one week of acclimation, the mice were intraperitoneally administered a single dose of AOM at 10 mg/kg (Sigma–Aldrich, Burlington, MA, USA). One week post-injection, a regimen of 2.5% (*w*/*v*) DSS (molecular weight 4000, Shanghai Macleans Biochemical Technology Co., Ltd., Shanghai, China) was introduced into the drinking water for one week. This cycle was repeated for three rounds, with each DSS exposure separated by a one-week interval of regular water supply. Following the final DSS cycle, mice were reverted to regular water until the conclusion of the experiment.

Dietary treatment was continuously given to mice through the entire experiment. The control group (CTL) was fed the AIN 93-G standard diet which containing casein, while the experimental group (FAA) received an isonitrogenous and isocaloric diet composed of free amino acids based on the AIN 93-G formulation. Both diets were provided by Jiangsu Synergy Pharmaceutical Bioengineering Co., Ltd. The diet composition is listed in Appendix A. For spontaneous genetically induced CRC models, male *Apc^Min/+^* mice at 6 weeks old were exposed to the same dietary treatment. The mice were euthanized at weeks 12 for AOM/DSS and *Apc^Min/+^* models.

### 2.2. Disease Activity Index (DAI)

The assessment was based on scoring three primary clinical signs. Weight loss was calculated as the difference between the initial and actual weights. Diarrhea was determined by evaluating the mucus and fecal matter adhering to the anal fur, and its consistency was confirmed based on the formation of fecal pellets. Rectal bleeding was identified using fecal occult blood test strips. Scores were assigned for the three primary clinical signs, namely percentage of weight loss, fecal consistency, and the presence of fecal blood. The macroscopic score was computed using the following formula: (weight loss score) + (diarrhea score) + (rectal bleeding score) [23].

### 2.3. Histopathological Analysis

Distal colon tissues were fixed in 10% neutral buffered formalin and subsequently embedded in paraffin. Tissue sections (5 μm) were deparaffinized in xylene, rehydrated through graded ethanol series, and then rinsed with water. For H&E staining, the sections were stained with hematoxylin, and counterstained with eosin. For PAS staining, the sections were treated with periodic acid solution for 10 min. Following a rinse with distilled water, the sections underwent Schiff staining and then hematoxylin counterstaining. For immunohistochemistry, after baking at 72 °C for 60 min, tissue sections were deparaffinized in xylene and rehydrated through an ethanol gradient. Endogenous peroxidase was quenched with 3% H_2_O_2_ for 30 min. Antigen retrieval was performed using 0.01 M citrate buffer with microwaving. Sections were blocked with 5% BSA. Primary antibodies Ki-67 (Abcam, ab16667, Waltham, MA, USA), ZO-1 (Abcam, ab276131, USA), E-cadherin (Abcam, ab231303, USA) were added and incubated at 4 °C overnight. After PBS wash, secondary antibodies were applied, followed by DAB development. Hematoxylin was used for counterstaining. After dehydration with alcohol, the sections were mounted and observed under a microscope with images captured for documentation.

### 2.4. Cell Culture and Drug Treatment

SW480 human colon cancer cell lines were cultured in Dulbecco’s Modified Eagle Medium (DMEM), supplemented with 10% fetal bovine serum (FBS) and 1% penicillin-streptomycin. Cultures were maintained in a humidified 5% CO_2_ atmosphere at 37 °C. Post 24-h adherence in 6-well plates, cells underwent treatment with DPA at concentrations of 0, 25, 50, and 100 μM for 48 h, to assess dose-dependent effects on cellular viability and morphology.

### 2.5. Quantitative Real-Time Polymerase Chain Reaction Analysis

Total RNA was isolated from the specimens utilizing TRIzol^TM^ reagent. Complementary DNA (cDNA) synthesis was conducted from these purified RNA samples employing the PrimeScriptTM RT reagent kit (Cat No. RR036A, TaKaRa, Japan). The quantitative real-time PCR (qRT-PCR) analyses were executed on the Light-Cycler 480 II system (Roche, Indianapolis, IN, USA) using SYBR^®^ Premix Ex TaqTM II (TaKaRa, Osaka, Japan), with primers obtained from the PrimerBank. GAPDH served as an internal normalization standard. Relative quantification of mRNA expression was calculated using the 2^−ΔΔCt^ method. The sequences of forward and reverse primers for the target genes are presented in Appendix A.

### 2.6. Multiplex Immunohistochemical (mIHC)

The sections initially underwent deparaffinization in xylene and a graded series of ethanol for rehydration. Antigen retrieval was achieved through microwave treatment in citrate buffer (pH 6.0) for 20 min. After cooling at room temperature, the sections were washed and subjected to a blocking step to prevent nonspecific binding, using a solution of 5% bovine serum albumin for 1 h. Primary antibodies, specifically Ki-67 (Clone MIB-1, Dako, Santa Clara, CA, USA) and EGLN 3 (SAB 55398-1, Santa Clara, CA, USA), were applied and incubated overnight at 4 °C. Following primary antibody incubation, sections were washed and incubated with fluorescently labeled secondary antibodies for 1 h at room temperature. To amplify the signals, appropriate fluorescent dyes were applied. Nuclear staining was performed using DAPI for 5 min. Finally, the stained sections were mounted using an anti-fade mounting medium. Fluorescence microscopic examination was conducted to analyze the expression patterns of the targeted proteins. Images were captured and quantitatively analyzed using an advanced image analysis software, enabling precise assessment of the staining intensity and distribution.

### 2.7. 16S rRNA Gene Sequencing

High-throughput sequencing was employed to analyze the cecal contents of Ctl and FAA mouse models. Upon the receipt of samples, total DNA was meticulously extracted using a standardized DNA extraction kit, ensuring optimal yield and purity. The extracted DNA underwent rigorous quality assessment, including quantification and purity checks via spectrophotometry.

Subsequent to quality verification, PCR amplification was performed, specifically targeting the V4 region of the bacterial 16S rRNA genes. This was achieved using the primers 515F and 806R, known for their high specificity to this region. Amplified products were then validated through gel electrophoresis to confirm the correct size and purity. Sequencing of the amplified products was conducted on the state-of-the-art Illumina NovaSeq 6000 system, provided by Novogene. This platform offers high-throughput sequencing capabilities, ensuring comprehensive coverage and depth. Post-sequencing, the raw data underwent processing to obtain clean tags using Vsearch V2.15.0, a robust tool for filtering and dereplicating sequence data. Further, tags matched to amplicon sequence variants (ASVs) were filtered and refined using Vsearch V2.4.4. For the taxonomic analysis, the refined sequences were aligned to the SILVA138 database, a comprehensive and curated database for microbial identification. The analysis of the taxonomic composition of the cecal microbiota was performed using QIIME 2 (version 2019.4.0), a widely recognized bioinformatics pipeline for microbiome analysis. This comprehensive approach allowed for a detailed understanding of the microbial diversity and structure in the cecal contents of the mouse groups under study.

### 2.8. Colon Transcriptome Profiling

RNA was meticulously extracted from distal colon tissues utilizing TRIzol reagent, a method known for its efficiency in isolating high-quality RNA. Following extraction, RNA purification was further enhanced using OligodT magnetic beads, which selectively bind to mRNA, ensuring a highly purified sample. The extracted RNA was then fragmented under controlled conditions to optimal sizes for cDNA synthesis.

For reverse transcription, fragmented RNA was converted into complementary DNA (cDNA) using a specific reverse transcriptase enzyme, under carefully optimized conditions to ensure high efficiency and fidelity. The resulting cDNA was then PCR-amplified using primers designed to anneal to regions of interest, ensuring the amplification of relevant gene sequences. Sequencing of the amplified cDNA was performed on the advanced DNBSEQ platform, a next-generation sequencing technology known for its high throughput and accuracy. The raw data generated were subjected to stringent quality control using SOAPnuke (v1.5.6) software, which filtered out low-quality reads and contaminants, ensuring only high-quality data were retained for analysis. The clean data obtained were aligned to the reference genome using HISAT and Bowtie2, two alignment tools that provide accurate and efficient mapping of sequencing reads. Gene expression levels were quantified using the RSEM (RNA-Seq by Expectation-Maximization) method, known for its accuracy in estimating gene and isoform abundance from RNA-Seq data. Gene annotation was carried out against comprehensive databases such as KEGG (Kyoto Encyclopedia of Genes and Genomes) and GO (Gene Ontology), providing insights into the biological functions and pathways associated with the expressed genes. Additionally, predictions of transcription factors were made, offering a deeper understanding of the regulatory mechanisms at play.

Differential gene expression analysis was conducted using robust statistical tools, DESeq and PossionDis, setting stringent criteria at a Fold Change of ≥2 and a *p*-value or false discovery rate (FDR) of ≤0.001. This approach ensured that only genes with significant expression changes were identified. Functional categorization of the differentially expressed genes was based on their GO and KEGG annotations, which facilitated the understanding of the biological implications of these gene expression changes. Enrichment analysis, executed using R’s phyper function and TermFinder, identified significantly enriched biological terms and pathways, considering a Q-value of ≤0.05 as indicative of significant enrichment. This comprehensive analysis provided a deep understanding of the molecular alterations and biological pathways influenced by the experimental conditions.

### 2.9. Metabolic Profiling of Colon Content

After thawing the samples on ice to preserve their integrity, they were treated with a precooled solution composed of methanol, acetonitrile, and water in a specific ratio (*v*/*v*/*v*), optimized for efficient metabolite extraction. The use of ultrasonication was employed for a precise duration at a controlled frequency, facilitating the disruption of cellular structures and enhancing the yield of metabolites. This was followed by centrifugation at a specific speed and temperature for an optimal time to ensure complete separation of the supernatant from the debris. The supernatant was then carefully vacuum dried under specific conditions to concentrate the metabolites without degrading them. For mass spectrometry analysis, the dried extract was reconstituted in a carefully measured volume of acetonitrile solution to achieve a consistent concentration across all samples. Chromatographic separation was achieved using the advanced Agilent 1290 Infinity LC UHPLC (Shanghai Applied Protein Technology Co., Ltd., CA, USA) system, equipped with both HILIC and C18 columns. The specific mobile phase compositions, along with their gradient elution protocols, were meticulously designed for the optimal resolution and separation of a wide range of metabolites. During the analysis, to preserve sample stability, all samples were maintained in a 4 °C autosampler. The implementation of quality control (QC) samples at regular intervals was a critical step to ensure the reliability and consistency of the data. The AB 6500+ QTRAP mass spectrometer was utilized for detection, with electrospray ionization (ESI) source conditions meticulously set for each analysis to ensure optimal ionization efficiency and sensitivity. The monitoring was performed in multiple reaction monitoring (MRM) mode, a technique chosen for its high specificity and sensitivity in quantifying targeted metabolites. The accompanying software was programmed to meticulously extract peak data, which was then used to calculate the content of each substance. The data analysis was comprehensive, encompassing not only statistical analysis to determine the significance of the findings but also differential metabolite screening to identify metabolites that showed significant changes in concentration. Finally, the Kyoto Encyclopedia of Genes and Genomes (KEGG) pathway analysis was employed to elucidate the biological pathways impacted by these metabolites, providing insights into their potential roles in the biological system under study.

## 3. Results

### 3.1. Free Amino Acid-Based Diet Attenuates the Tumor Progression in AOM/DSS-Induced Colorectal Cancer Mouse Model

To evaluate the impact of protein digestibility on murine colorectal cancer, we fed mice with a diet composed of FAA and a nitrogen-balanced diet containing whole casein (CTL) for twelve weeks, (n = 5 pre group). Subsequently, the mice were treated with a single dose of azoxymethane (AOM) followed by three cycles of dextran sulfate sodium (DSS) (Figure 1A). Colonic shortening and histopathological alterations served as the primary indicators of colorectal carcinogenesis. Mice fed with CTL diet experienced premature mortality during the experimental period compared to those fed with FAA (Figure 1B). Despite having a higher body weight at week 12 with no statistical significance (Figure 1C), the CTL group exhibited colonic shortening compared to those of the FAA group (Figure 1D). Histological assessments by hematoxylin and eosin (H&E) staining revealed disruptions in the intestinal mucosal barrier, cellular heterogeneity, and tumor infiltration depth after AOM/DSS treatment (Figure 1E). Notably, there was less irregular and multilayered epithelial cell arrangement in the colon tissues of the mice fed with the whole casein diet compared with mice receiving an FAA diet. Periodic acid–Schiff (PAS) staining revealed that the FAA significantly promoted mucin secretion from goblet cells. Furthermore, immunohistochemical (IHC) staining of mouse colon tissues utilizing markers including Ki-67, E-Cadherin, and ZO-1 indicated that the FAA decreased cell proliferation and improved the tight junction integrity of CRC (Figure 1E). Based on these findings, it was evident that the FAA may alleviate tumor development in the carcinogen-induced CRC model.

### 3.2. Free Amino Acid-Based Diet Modulates Gut Microbiota in AOM/DSS-Induced Colorectal Cancer Mouse Model

To investigate the influence of dietary composition on gut microbiota, we performed 16s rRNA gene sequencing on cecal contents of male Balb/c mice subjected to AOM/DSS treatment under different dietary regimens. Upon computation and comparison of alpha diversity indices, we found that mice on the FAA diet exhibited reduced species richness compared to the control group, whereas did not reach statistical significance (Figure 2A). Principal Coordinate Analysis (PCoA) revealed a distinct reshaping of the gut microbial community with the amino acid regimen, with significant alterations observed at the species level (Figure 2B). Notably, the FAA diet significantly enriched beneficial bacteria including Akkermansia, Bacteroides, Lactobacillus, and Parabacteroides (Figure 2C,D). In contrast, an elevated abundance of Blautia was detected in the CTL group. Furthermore, microbial network analysis was performed for the two groups, revealing co-exclusivity relationships between probiotics and pathogenic bacteria in the FAA group, notably between Akkermansia and Coriobacteriaceae UCG-002 (Figure 2E). These data suggest that an FAA might reverse microbial dysbiosis associated with CRC.

### 3.3. Free Amino Acid-Based Dietary Intervention Attenuates Tumor Progression in Spontaneous Genetically Induced Colorectal Cancer Mouse Model

To further delineate the direct association between the FAA diet and CRC progression, we employed the spontaneous genetically induced CRC mouse model (Apc^Min/+^) for validation, conducting comprehensive sequencing analyses encompassing 16s rRNA gene, transcriptomics, and metabolomics (Figure 3A), (n = 5 pre group). In terms of body weight and colon length, mice administered with a whole casein diet mirrored the phenotype of AOM/DSS-treated Balb/c mice (Figure 3B,D), exhibiting reduced colon length and a progressive elevation in DAI (Figure 3C). These mice manifested severe diarrhea, hematochezia, and a notable increase in intestinal tumors compared to the amino acid-based diet cohort (Figure 3E). Consistently, Periodic acid–Schiff (PAS) and hematoxylin–eosin (H&E) staining, along with immunohistochemical staining for ki-67, E-Cadherin, and ZO-1, corroborated the findings observed in the chemically-induced model (Figure 3F). Collectively, these data confirmed that the FAA may mitigate the progression of spontaneous genetically induced CRC in mice.

### 3.4. Free Amino Acid-Based Dietary Intervention Revert Gut Microbiota Dysbiosis in Spontaneous Genetically Induced Colorectal Cancer Mouse Model

Upon analyzing the 16s rRNA gene sequencing of the cecal contents in mice, a significant reduction in alpha diversity was observed in the group treated with the amino acid-based diet (Figure 4A). Principal Coordinates Analysis (PCoA) highlighted a pronounced distinction between the two groups (Figure 4B). At the genus level, the experimental group predominantly exhibited an upregulation of beneficial bacteria, with a notable enrichment of Akkermansia and Bacteroides (Figure 4C). In contrast, the control group was characterized by an upregulation of potential pathogens, including UCG-014 and Odoribacter (Figure 4D,E). Ecological network analysis, akin to findings in the DSS/AOM model group, revealed co-exclusion correlations between beneficial and pathogenic bacteria, encompassing associations between Akkermansia and UCG-014, as well as Bacteroides (Figure 4F). These consistent findings suggest that the FAA may prevent the progression of CRC through modulating gut microbiota.

### 3.5. Free Amino Acid-Based Dietary Intervention Modulate Intestinal Metabolites in Spontaneous Genetically Induced Colorectal Cancer Mouse Model

Given the significant role of metabolites secreted by the gut microbiota in regulating health and disease, we performed a metabolomic analysis on the cecal contents in the CTL and FAA-fed Apc^Min/+^ mice. The OPLS-DA supervised analysis indicated a clear distinction between the two groups (Figure 5A). A total of 46 metabolites exhibited significant alterations, with a predominant upregulation in the FAA-fed Apc^Min/+^ mice compared with CTL-fed Apc^Min/+^ mice (Figure 5B). This upregulation primarily involved metabolites from the ornithine cycle and an increase in fatty acid metabolites, such as DPA, α-linolenic acid, and γ-linolenic acid (Figure 5C). To associate these metabolites with potential metabolic activities of the gut microbiota, we conducted a correlation analysis of the altered bacteria and metabolites. We found that Akkermansia, Bacteroides, and Lactobacillus positively correlated with long-chain polyunsaturated fatty acids, including DPA, Elaidic acid, Docosatetraenoic acid, and linolenic acid. Our analysis revealed that the FAA altered gut microbiota and their associated metabolites might contribute to CRC suppression.

### 3.6. Free Amino Acid-Based Dietary Intervention Suppresses Carcinogenic Signaling in Spontaneous Genetically Induced Colorectal Cancer Mouse Model

To elucidate the molecular mechanisms underlying the suppressive effects of FAA on colorectal cancer (CRC), transcriptomic sequencing was performed on distal colon tissues of Apc^Min/+^ mice. Compared to the FAA-treated group, the CTL group exhibited upregulation of 184 genes and downregulation of 127 genes (Figure 6A), as detailed in Appendix A. Notably, genes that were downregulated predominantly clustered in cancer-related pathways, including Hippo, mTOR, and Wnt. Genes implicated in cell proliferation and migration such as Tgfβ3, Fzd2, Fzd9, and Lpar3 showed marked downregulation in tumor tissues post-FAA treatment (Figure 6B). Pathway enrichment analysis of differentially expressed genes and metabolites revealed common enrichment in pathways like mTOR, PI3K-Akt, and FoxO signaling (Figure 6C). Furthermore, significant upregulation of Egln3, an enzyme linked to cellular oxygen sensing, was observed in FAA-treated mouse colon samples. This upregulation may influence the stability of HIF-1α under hypoxic conditions. Analysis of the TCGA database demonstrated a correlation between increased EGLN 3 expression and enhanced post progression survival (PPS) (Figure 6D). The effect of DPA on EGLN3 expression was validated in SW480 colorectal cancer cells, revealing a concentration-dependent upregulation in response to DPA (Figure 6E). Differential metabolite correlation analysis further confirmed a significant positive correlation between EGLN 3 and DPA (Figure 6F). Multicolor immunofluorescence histochemistry on colon sections from the FAA group corroborated the heightened expression of EGLN 3 and a concurrent reduction in Ki-67 expression (Figure 6G). Together, these preliminary findings provide new insights into the molecular mechanisms of an FAA-based diet in attenuating CRC and may suggest novel therapeutic targets for future strategies.

## 4. Discussion

Recent research highlights the complex relationship between diet, gut microbiota, and colorectal cancer (CRC) progression [24]. Our study reveals the FAA’s significant role in affecting CRC progression and gut microbiota in murine models, emphasizing the protective role of amino acid-based diets. We observed colonic shortening and histopathological changes in mice on a casein-complete diet, indicating the adverse impact of diet on CRC. Importantly, dietary interventions showed marked effects on tight junction integrity and cell proliferation in AOM/DSS and *Apc*^Min/+^ mouse models, underscoring their importance in CRC management. Our findings suggest that amino acid-rich diets bolster digestibility and support the integrity of the intestinal barrier, thereby mitigating the adverse impacts of unabsorbed dietary proteins on colonic shortening. The 16s rRNA gene sequencing analysis revealed distinct microbial communities in amino acid diet and control groups, with the FAA-treated group showing increased beneficial bacteria like *Akkermansia* and *Bacteroides*, contrasting with pathogen rise in the control. An ecological network analysis suggests the amino acid diet reshapes gut microbiota, potentially enhancing beneficial communities and reducing CRC progression.

As pivotal mediators of interactions between gut microbiota and the host, gut metabolites play a crucial role in host health and disease [25]. Here, we found the FAA fed group exhibited significant alterations in 46 metabolites, predominantly up-regulating metabolites associated with the ornithine cycle and specific fatty acid metabolites, such as DPA, α-linolenic acid, and γ-linolenic acid [26]. Numerous epidemiological studies have identified that n-3 long-chain polyunsaturated fatty acids can mitigate the onset and progression of malignant tumors in humans [27]. Further correlation analysis also identified a positive association between beneficial bacteria like *Akkermansia*, *Bacteroides*, and *Lactobacillus* with long-chain polyunsaturated fatty acids, suggesting a mutual support mechanism [28]. These findings imply that the synergistic effects of specific gut metabolites and microbial communities might enhance the chemopreventive effects of the amino acid-based diet against CRC [29].

Microbial entities associated with CRC have been demonstrated to instigate various inflammatory and carcinogenic pathways. Given that the amino acid-based diet can promote beneficial gut metabolites and restore gut barrier functions in CRC models, we delved deeper into the molecular antitumor mechanisms of the amino acid diet in CRC. Transcriptomic features of distal colon tissues revealed potential impacts of the amino acid-based diet on cancer-associated pathways. A salient observation was the downregulation of genes in cancer-associated pathways, such as Hippo, mTOR, and Wnt, including genes like tgfb3, fzd2, fzd9, and lpar3, which are closely associated with cell proliferation and migration, underscoring the potential therapeutic implications of the amino acid-based diet [30]. The roles of fzd2 and fzd9, core components of the Wnt signaling pathway, in CRC pathogenesis further accentuate the significance of our findings. Another intriguing observation was the upregulation of EGLN 3, an enzyme associated with cellular oxygen sensing, which might influence the stability of HIF-1α under hypoxic conditions [31]. The association between EGLN 3 upregulation and patient survival rates derived from the TCGA database suggests a potential prognostic role of EGLN 3 in CRC. Our differential metabolite correlation analysis reveals a positive correlation between EGLN 3 and DPA, offers a novel avenue to elucidate the interplay between metabolic and genetic factors in CRC progression [32]. Thus, in line with our hypothesis, the amino acid-rich diets not only support digestive health and intestinal barrier integrity but also contribute to a reduction in the effects of unabsorbed dietary proteins on colonic shortening, providing a holistic approach to CRC management and prevention. However, while our study provides novel insights into the molecular intricacies of CRC under an amino acid-based diet backdrop, these findings should be approached with caution. It is important to note the limitation regarding the lack of baseline data for cecal contents before dietary intervention. This omission could potentially influence the interpretation of our results, as baseline conditions are crucial for understanding the full impact of dietary changes. Rigorous further investigations are imperative to validate these preliminary observations and explore their translational potential in clinical settings.

## 5. Conclusions

This study demonstrates that a diet based on free amino acids can alter the composition of the gut microbiota, increase beneficial bacteria, promote the secretion of protective metabolites, enhance gut barrier functions, and inhibit carcinogenic pathways. It provides new insights into the potential of free amino acid-based diets in combating colorectal cancer (CRC) and lays the foundation for future dietary adjustments for the prevention and treatment of CRC.

However, applying these research findings in practice requires personalized dietary recommendations, taking into account the variability in human responses to diet. Further research is needed to understand how such a diet specifically affects CRC, including its interactions with gut microbiota and metabolites.

Future efforts should explore how to integrate dietary interventions with existing treatments and enhance interdisciplinary collaboration to better understand the relationships between diet, gut microbiota, and CRC. This will aid in developing more effective dietary strategies for the prevention and treatment of CRC.

## Figures and Tables

**Figure 1 nutrients-16-01040-f001:**
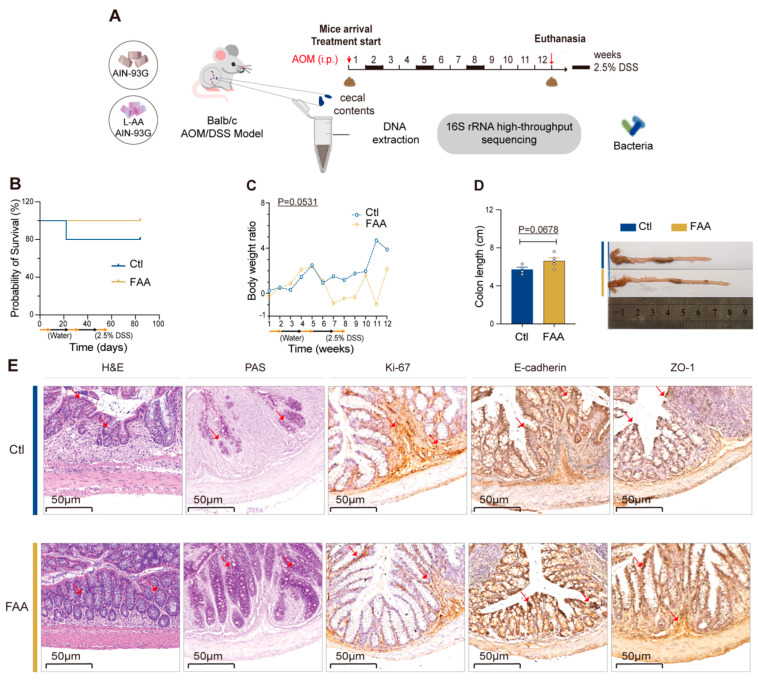
Free amino acid-based diet attenuates CRC development in an AOM/DSS mouse model. (**A**) Illustrates a detailed schematic of the experimental design, highlighting the use of an FAA diet in the AOM/DSS-induced CRC mouse model, depicting the time points of diet administration, AOM/DSS treatment, and subsequent analysis phases (n = 5 per group). (**B**,**C**) Present Kaplan–Meier survival curves, showcasing the survival rates over time in the AOM/DSS-treated mice, along with longitudinal body weight changes, providing insights into the health status and progression of the disease in the mice under different dietary conditions (n = 4–5 per group) survival curves analyzed by log-rank test. (**D**) Features a quantitative assessment of colon lengths in both control CTL and FAA diet groups (n = 4–5 per group). Date by unpaired *t* test. This is further enriched with representative macroscopic images of the colons, allowing for visual comparison between groups. (**E**) Displays Representative histological and immunohistochemical evaluations, including H&E and PAS staining, alongside immunostaining for Ki-67, E-cadherin, and ZO-1 in colon tissues. Notable pathological changes and protein expression patterns are indicated with red arrows, highlighting areas of interest. Scale bar represents 50 μm, providing a reference for tissue structure size.

**Figure 2 nutrients-16-01040-f002:**
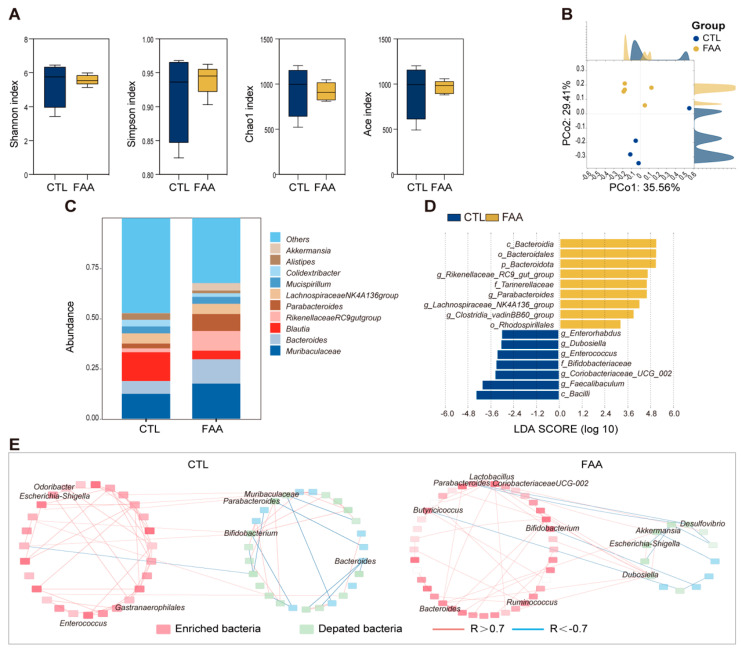
Effects of a free amino acid-based diet on gut microbiota in an AOM/DSS mouse model. (**A**) Depicts alpha diversity indices, including Shannon, Simpson, Chao1, and ACE metrics. These indices provide insights into the microbial richness and evenness in the studied samples, facilitating a deeper understanding of microbial community complexity (n = 4–5 per group). (**B**) Illustrates a PCoA plot, based on unweighted UniFrac distances (n = 4–5 per group). (**C**) Shows a bar chart detailing the relative abundance of the top ten microbial species identified in the samples (n = 4–5 per group). This visualization allows for a straightforward comparison of predominant species between different groups or conditions. (**D**) Presents the results from Linear Discriminant Analysis (LDA), a method used to identify features (in this case, microbial taxa) that most effectively differentiate between groups, with the LDA score indicating the effect size. (**E**) Features an ecological network analysis, offering a graphical representation of the interactions within the microbial community (n = 4–5 per group). Correlations were statistically analyzed using SPSS (IBM 27.0) software, and only interactions with strength differences of ≥0.7 or ≤−0.7 were considered significant and visualized. In the network, blue lines represent negative correlations, while red lines denote positive correlations, providing a clear depiction of the ecological relationships. This comprehensive suite of analyses collectively informs on the diversity, structure, and interrelationships of the microbial community under investigation, employing a range of statistical and graphical methods to elucidate complex ecological dynamics.

**Figure 3 nutrients-16-01040-f003:**
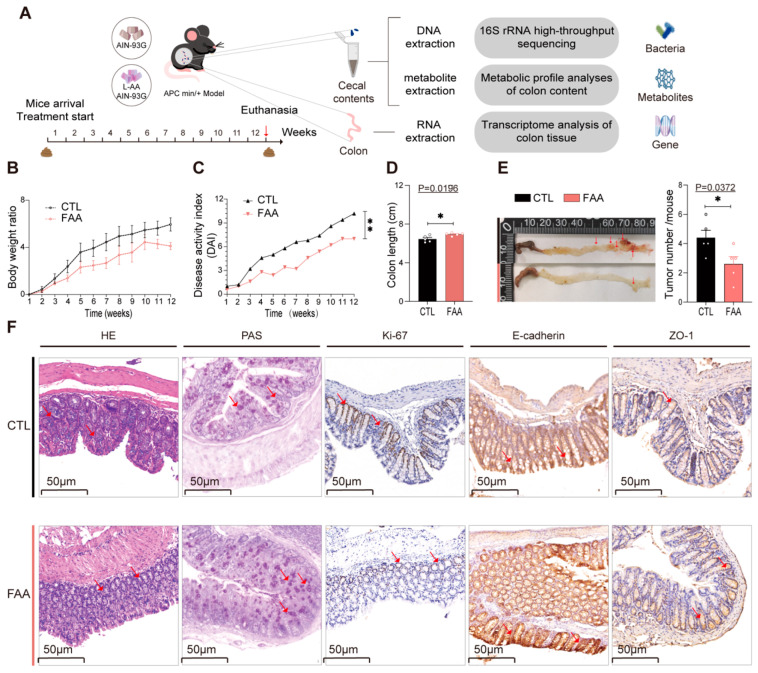
Free amino acid-based diet attenuates CRC development in an *Apc^Min/+^* mouse model. (**A**) Outlines the experimental design, which integrates a nutritional intervention with comprehensive molecular analyses. The diagram details the sequential steps including administration of a free amino acid-based diet, followed by 16s rRNA gene sequencing of cecal contents, metabolomic profiling, and transcriptomic analysis of colonic tissues, providing a holistic view of the dietary impact at various biological levels (n = 4 per group). (**B**) Tracks body weight changes over the course of the experiment, reflecting the general health and response to the diet in mice (n = 4 per group). (**C**) Depicts the Disease Activity Index (DAI), quantifying inflammation levels, crucial for assessing the severity of colitis (n = 4 per group), survival curves analyzed by log-rank test. (**D**) Provides measurements of colonic length, a parameter inversely correlated with inflammation severity (n = 4 per group). (**E**) Illustrates colonic tumor metrics, complemented by representative images showcasing tumor counts, offering a visual and quantitative assessment of tumorigenesis (n = 4 per group). (**F**) Displays Representative histological examinations using hematoxylin and eosin (H&E) staining, Periodic acid–Schiff (PAS) staining, and immunohistochemical staining for markers Ki-67, E-cadherin, and ZO-1. Positive markers are indicated with red arrows, highlighting areas of interest. Data are presented as mean ± SEM, providing statistical rigor. Significance is denoted as *, *p* < 0.05; **, *p* < 0.001, determined by Student’s *t*-test, lending credibility to the observed differences. This figure amalgamates various investigative approaches, from dietary intervention to molecular and histological analyses, to provide a comprehensive understanding of the dietary effects in this genetically predisposed CRC model.

**Figure 4 nutrients-16-01040-f004:**
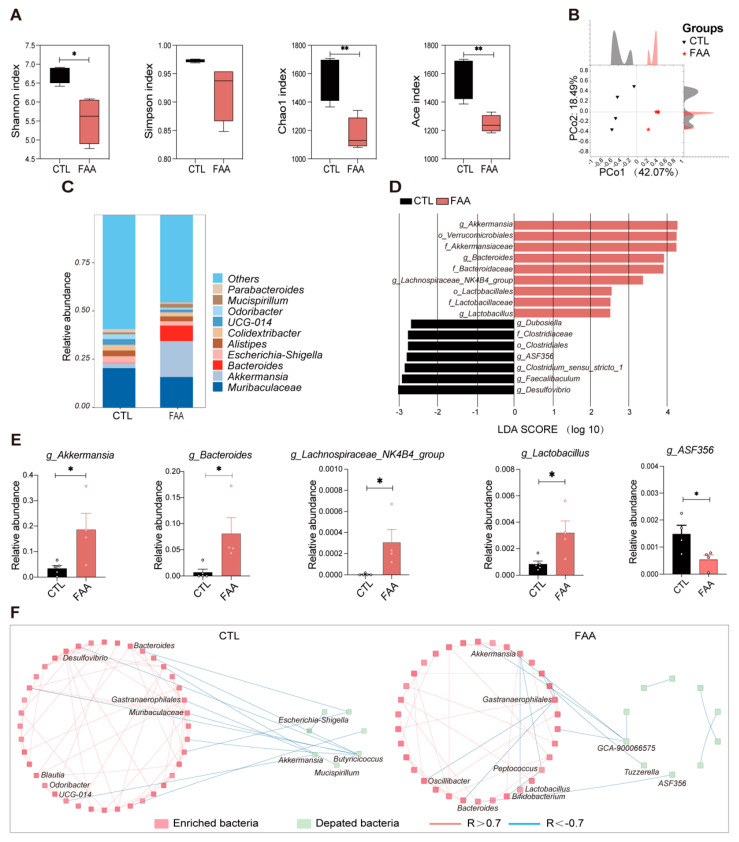
Free amino acid-Based diet reverses gut microbiota dysbiosis in an *Apc^Min/+^* mouse model. (**A**) Alpha diversity indices (Shannon, Simpson, Chao1, ACE) of gut microbiota (n = 4 per group). (**B**) Principal Coordinates Analysis (PCoA) illustrating beta diversity (n = 4 per group). (**C**) Bar chart depicting the relative abundance of the top ten microbial species (n = 4 per group). (**D**) Results from Linear Discriminant Analysis (LDA) (n = 4 per group). (**E**) Statistical analysis of differential bacteria between the experimental and control groups (n = 4 per group). (**F**) Ecological network analysis showcasing interactions between differential bacteria in CTL and FAA groups (n = 4 per group). Correlations were statistically analyzed using SPSS. Only interactions with strength differences of ≥0.7 or ≤−0.7 between groups were visualized. Blue lines represent negative correlations, while red lines indicate positive correlations. Significance is denoted as *, *p* < 0.05; **, *p* < 0.01, determined by Student’s *t*-test, lending credibility to the observed differences.

**Figure 5 nutrients-16-01040-f005:**
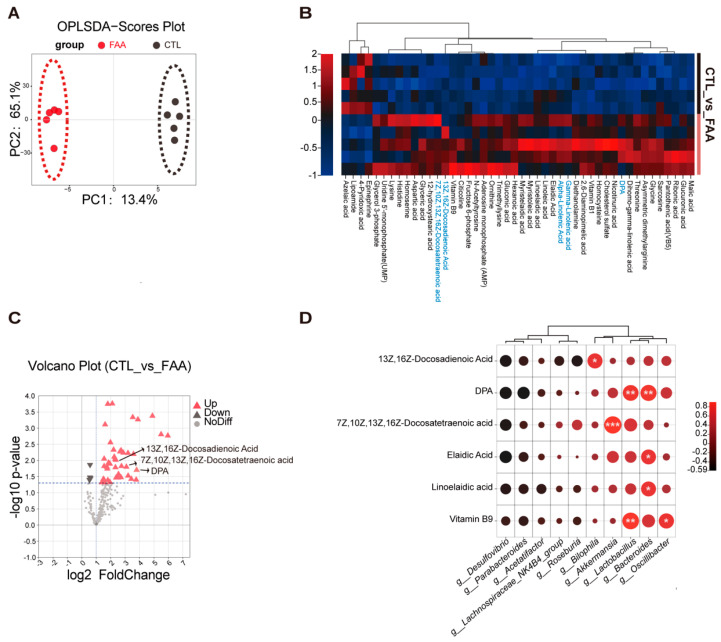
Free amino acid-based diet modifies cecal microbiota-associated metabolites in an *Apc^Min/+^* model mice. (**A**) Displays results from orthogonal partial least squares discriminant analysis (OPLS-DA), indicating a clear separation of metabolic profiles. This separation suggests distinct metabolic changes between the studied groups, highlighting the impact of the dietary intervention (n = 5 per group). (**B**) Showcases a differential metabolite heatmap, revealing several upregulated metabolites in the FAA group (n = 5 per group), including Docosapentaenoic acid (DPA), γ-linolenic acid, and other fatty acids. This heatmap provides a visual representation of the relative abundance of these metabolites, illustrating significant metabolic shifts due to the dietary treatment. (**C**) Depicts a volcano plot, highlighting numerous upregulated differential metabolites, including DPA. This plot combines statistical significance with the magnitude of change, enabling the identification of key metabolites altered by the FAA diet (n = 5 per group). (**D**) Presents a Pearson correlation map, demonstrating a positive correlation between DPA and specific microbial genera such as *Akkermansia*, *Bifidobacterium*, and *Lactobacillus*. This correlation analysis suggests potential interactions between specific dietary components and gut microbiota, providing insights into the underlying mechanisms of dietary effects. Significance is denoted as *, *p* < 0.05; **, *p* < 0.01, ***, *p* < 0.001, determined by Student’s *t*-test, lending credibility to the observed differences.

**Figure 6 nutrients-16-01040-f006:**
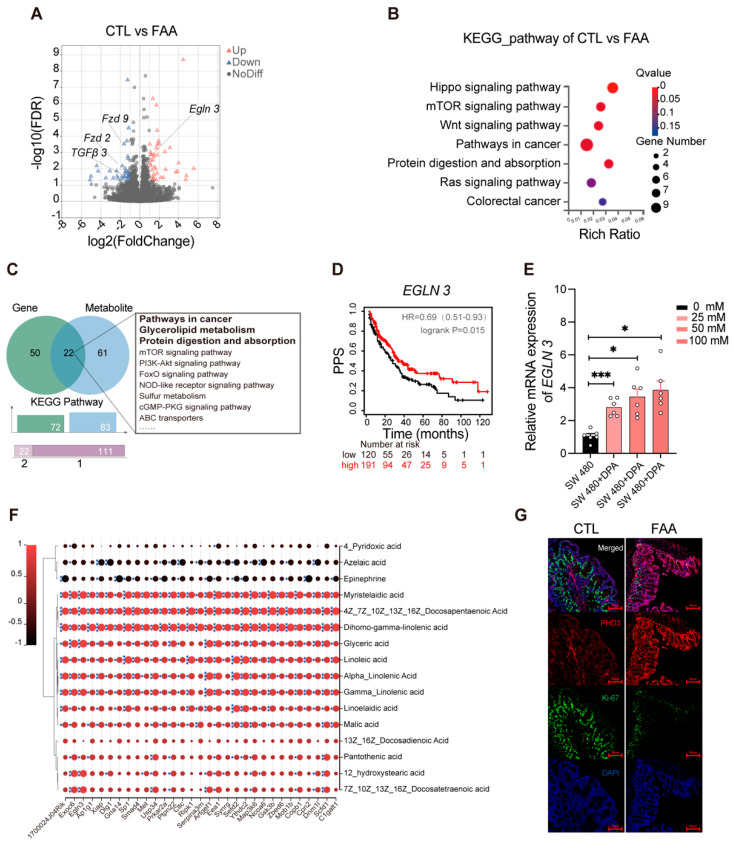
Free Amino Acid-Based Diet inhibits carcinogenic signaling in colon tissues of *Apc^Min/+^* mice. (**A**) Presents a volcano plot illustrating upregulation of multiple genes in the FAA group, notably including Egln3 (n = 5 per group). (**B**) Demonstrates pathway enrichment analysis for downregulated differential genes, predominantly in the Hippo, mTOR, and Wnt signaling pathways (n = 5 per group). (**C**) Displays a Venn diagram of enriched pathways from differential genes and differential metabolites, providing a comprehensive view of the interconnected pathways affected by the FAA diet at both the genetic and metabolic levels. (**D**) Kaplan–Meier plotter, indicating that upregulation of the EGLN 3 gene is associated with increased PPS, with a statistical significance of *p* = 0.015. (**E**) Reports the expression levels of the target gene EGLN 3 in cells post 48-h treatment with DPA, as determined by quantitative PCR (qPCR) analysis. (**F**) The correlation between differential metabolites and differential genes. The horizontal axis represents differential genes, and the vertical axis represents differential metabolites. (**G**) Features a multicolor immunohistochemical fluorescence image, with Egln3 stained in red, Ki-67 in green, and nuclei counterstained with DAPI in blue. Significance is denoted as *, *p* < 0.05; ***, *p* < 0.001, determined by Student’s *t*-test, lending credibility to the observed differences.

## Data Availability

The raw data of 16S rRNA gene sequencing generated in this study have been deposited at NIH National Center for Biotechnology Information (accession number: PRJNA1036134).

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
