# Peer review of "A Free Amino Acid Diet Alleviates Colorectal Tumorigenesis through Modulating Gut Microbiota and Metabolites"

_nutrients, 2024, doi:10.3390/nu16071040_

Round 1

Reviewer 1 Report

Comments and Suggestions for Authors

This study investigates the impact of dietary protein digestibility on colorectal cancer (CRC) using mouse models. A free amino acid (FAA) diet is compared to a casein protein diet (CTL), revealing that FAA significantly attenuates CRC progression, enriches beneficial gut bacteria, alters metabolites, and regulates cancer-associated pathways, indicating potential therapeutic benefits in inhibiting carcinogenesis.

This is a very well written manuscript and the researches are well performed.

However several points should be addressed.

Free amino acid meal should be described in detail

Absorption ratio of the free amino acids in the small intestine should be described.

Some, if any, adverse effects of aromatic amino acids should be described.

Comments on the Quality of English Language

no problem

Reviewer 2 Report

Comments and Suggestions for Authors

The investigators report on the effects of a casein protein diet (CTL) versus a free amino acid (FAA)-based diet on CRC progression in two standardized mouse models to evaluate both carcinogen- induced and genetic risk as causes of colon cancer compared to controls. The study objective addresses important research questions pertinent to current increased incidence of CRC in younger persons. The investigators have undertaken an in-depth study using a comprehensive approach. The experimental design has been rigorously carried out. 

The study methods are provided in detail. Results are clearly described and accompanied by appropriate figures and analysis. Studies of metabolic shift in cecal microbiota in response to diet are novel. One general concern is the lack of information about the numbers of mice in experiments and in data available for analysis and conclusions. Therefore the overall significance of the report will require additional study and further confirmation. Despite these concerns, the investigation as reflected in the current report is highly interesting, informative, and can make a significant contribution to the field. 

Specific comments and questions:

1.     Effect of FAA on tumor progression in AOM/DSS treated mice: The investigators report that some mice on whole casein (CTL) diet died before 3 weeks in the experimental period with AOM/DSS despite having a higher weight. Figure 1 B depicted in days, shows a sharp drop due to death just before 3 weeks while Fig 1 C shows data in weeks that survivors gained weight after this point. Additional information regarding cause of death, food and water consumption, environmental differences that might explain the drop-off are needed. The differences in colon length in Fig 1D are modestly appreciable. Although selected images (Fig 1E) support significant descriptive differences, information about the source- whether or not each image came from the same or different mice in the two cohorts would be helpful.

2.     FAA modulation of microbiota in AOM/DSS model:  The results of FAA on 16s sequencing shown in Fig 2 indicate reduced diversity in FAA mice (2A),  If baseline data in this group can confirm change due to diet and then be compared to same to data from CTL treated mice, the conclusion can be verified.  To support possible reshaping differences shown in Fig 2B, numbers of mice in groups should be provided.

3.     Effect of FAA on tumor after AOM/DSS treatment in spontaneous (Apcmin/+ mice) mouse model:  Fig. 3 shows some reduction in colon length while number of tumors is strikingly increased under CTL diet that was sharply reduced with FAA diet (3E). Progressive elevation in Disease Activity Index (DAI) (Fig. 3C) was not shown for the AOM/DSS model as implied in text of Results. DAI was slightly lower with CTL compared to FAA diet in the spontaneous model and appears to have the same slope indicating similar rate of change under both diets in the spontaneous genetic model.  

4.     Metabolomic analysis on the cecal contents after dietary intervention in Apc min/+ mice.  Add information about sample size and baseline contents. 

5.     The study hypothesis is that amino acid-rich diets improve digestibility supporting the integrity of the intestinal barrier and reducing the effects of unabsorbed dietary proteins on colonic shortening could be better integrated in the discussion. 

Reviewer 3 Report

Comments and Suggestions for Authors

Yang-Meng Yu et al. submitted an original article entitled  „A Free Amino Acid Diet Alleviates Colorectal Tumorigenesis Through Modulating Gut Microbiota and Metabolites”. Colorectal cancer has become a global health problem worldwide, especially in developed countries in Europe and North America. It is categorized as a disease of the middle-aged and elderly. However, unfortunately, this problem concerns younger and younger people. The progression of colorectal cancer often comprises a sequential transformation from normal mucosa to adenoma, resulting in malignant transformations. Thus, early prevention is paramount.
The authors aim „to elucidate how free amino acid-based diets influence the gut microbiota and their impacts on the genetic and metabolic levels”. They discovered the potential interactions between gut microbiota and metabolic products, leading to the inhibition of specific pathways, thereby decelerating progression. In this order, they employed complex methodology, including rRNA gene sequencing, transcriptomics, metabolomics, and immunohistochemistry. The authors observed that free amino acids attenuated the progression of colorectal cancer in a significant way. In conclusion, they state that free amino acid-based diets modulate gut microbial composition, enhance protective metabolites, improve gut barrier functions, and inhibit carcinogenic pathways.
This is a very comprehensive project described in detail. It is some kind of novelty in the field and could stimulate further advanced studies in the effective and safe strategies of cancer management. The manuscript is well prepared either from the substantive or technical side. It is well suited to the profile of Nutrients and could be interesting for the readers.
I have only minor comments:
Conclusions could be extended. This part does not include all aspects of extensive research. Authors could describe future perspectives and focus on the challenges/problems more precisely.
A lot of data are presented in the paper – maybe some of the tables could be transported to the supplementary materials? The same applies to drawings (and, labels are sometimes too small).
By the way, tables S1-S3 are included in the manuscript (Table „S”… could be rather in the Supplementary Material). I do not see Table 3 (line 139)…
Ref. no 23 is absent in the text – authors should introduce it. The manner of the presentation of the reference should be corrected according to the Guides for Authors.
